# Overexpression of *BmJHBPd2* Repressed Silk Synthesis by Inhibiting the JH/Kr-h1 Signaling Pathway in *Bombyx mori*

**DOI:** 10.3390/ijms241612650

**Published:** 2023-08-10

**Authors:** Jikailang Zhang, Xia Zhang, Hui Zhang, Jiaojiao Li, Wei Li, Chun Liu

**Affiliations:** 1State Key Laboratory of Resource Insects, Southwest University, Chongqing 400715, China; zhangjikailang@163.com (J.Z.);; 2Jinfeng Laboratory, Chongqing 401329, China

**Keywords:** juvenile hormone, juvenile hormone binding protein, silk protein, *Bombyx mori*

## Abstract

The efficient production of silkworm silk is crucial to the silk industry. Silk protein synthesis is regulated by the juvenile hormone (JH) and 20-Hydroxyecdysone (20E). Therefore, the genetic regulation of silk production is a priority. JH binding protein (JHBP) transports JH from the hemolymph to target organs and cells and protects it. In a previous study, we identified 41 genes containing a JHBP domain in the *Bombyx mori* genome. Only one JHBP gene, *BmJHBPd2*, is highly expressed in the posterior silk gland (PSG), and its function remains unknown. In the present study, we investigated the expression levels of *BmJHBPd2* and the major silk protein genes in the high-silk-producing practical strain *872* (*S872*) and the low-silk-producing local strain *Dazao*. We found that *BmJHBPd2* was more highly expressed in *S872* than in the *Dazao* strain, which is consistent with the expression pattern of fibroin genes. A subcellular localization assay indicated that *BmJHBPd2* is located in the cytoplasm. In vitro hormone induction experiments showed that *BmJHBPd2* was upregulated by juvenile hormone analogue (JHA) treatment. *BmKr-h1* upregulation was significantly inhibited by the overexpression of *BmJHBPd2* (BmJHBPd2OE) at the cell level when induced by JHA. However, overexpression of *BmJHBPd2* in the PSG by transgenic methods led to the inhibition of silk fibroin gene expression, resulting in a reduction in silk yield. Further investigation showed that in the transgenic BmJHBPd2OE silkworm, the key transcription factor of the JH signaling pathway, Krüppel homolog 1 (Kr-h1), was inhibited, and 20E signaling pathway genes, such as broad complex (Brc), E74A, and ultraspiracle protein (USP), were upregulated. Our results indicate that *BmJHBPd2* plays an important role in the JH signaling pathway and is important for silk protein synthesis. Furthermore, our findings help to elucidate the mechanisms by which JH regulates silk protein synthesis.

## 1. Introduction

The juvenile hormone (JH) plays a key role in insect development, metamorphosis, and reproduction [1,2]. Juvenile hormone binding protein (JHBP) is the first key factor in the JH signaling pathway. The JHBP or JH-JHBP complex is a vital member of the intricate JH signaling transmission chain that binds to membrane receptors [3,4]. It broadens our understanding of the crucial factors involved in the JH signaling pathway and its mode of action in target organs and has far-reaching importance in pest control and other practical applications.

JH is a hemiterpenoid compound secreted by the corpora allata [5]. JHBP is a carrier protein that works in the hemolymph and cells. This protein transports hormone molecules from the JH synthesis site in the corpora allata to the peripheral target cells and assists JH molecules in entering the circulatory system. JHBP can also reduce the non-specific binding of JH [6], protecting it from enzymatic degradation [7,8]. Intracellular JHBP can be divided into cytoplasmic and nuclear JHBP, which are both involved in the effects of JH on target genes. However, to date, the progress of related research has been relatively slow [9,10]. The expression level of JHBP, which was found to be predominant in the fat body of the bamboo borer, was average from the third to the fifth stages. The expression level was the highest at the early diapause stage, which continued until the middle diapause stage and then decreased until the pupae stage [11]. In melon aphids, RNA interference has been used to silence the expression of JHBP and block the transmission of JH signals, resulting in mortality and thus allowing effective pest management [12]. Proteomic analysis of male accessory gland secretions has shown that JHBPs affect female reproduction in oriental fruit flies [13]. Therefore, these studies prove that JHBP is involved in metabolism, growth, and reproduction. The JHBPs from this gene family are found in many lepidopteran species and form a separate group from other genes [14,15]. To date, studies on the JHBP have focused on many lepidopteran insects, including the tobacco hornworm (*Manduca sexta*) [16,17], tobacco budworm (*Heliothis virescen*) [7,18], greater wax moth (*Galleria mellonella*) [19], and silkworm (*Bombyx mori*) [20,21].

The domesticated silkworm is a typical lepidopteran holometabolous insect model organism. As a natural protein fiber, the use of silk has been applied in many fields [22]. Silkworm silk glands primarily synthesize and secrete two types of silk proteins, namely fibroin and sericin. Fibroin is composed of a heavy (Fib-H) and light (Fib-L) chain, and glycoprotein 25 kDa (P25) is synthesized in the posterior silk gland (PSG) [23,24]. Previous studies have found that increasing the amount of silk protein, injection, smearing, and feeding on exogenous JH or JH analogs (JHAs) can prolong the larval stage of silkworms and increase the RNA and DNA contents of the silk protein gene [25,26,27,28]. The mechanism of action behind this remained unclear until recent results showed that Krüppel homolog 1 (Kr-h1) is involved in the repression of metamorphosis. In transgenic silkworms with *Kr-h1* overexpression, the silk glands were significantly enlarged [29]. Furthermore, JH induces extended expression of the *Bmfib-H* gene, and *BmKr-h1* may suppress larvae–pupae metamorphosis by activating the expression of *B. mori*-derived dimmed (*Bmdimm*). *Bmdimm* is a transcription factor involved in the regulation of silk gene transcription that activates the transcription of the *Bmfib-H* gene in *B. mori* posterior silk gland (PSG) cells [30].

In previous studies, 41 JHBP genes were identified in silkworms, which contained conserved structures of the binding proteins of JH. Microarray data have shown different JHBP gene expression trends in silkworms. The expression of *BmJHBP* genes was generally higher in the head, integument, midgut, fat body, testes, and ovaries. Expression levels also differed among different tissues. The PSG had a specific and high expression of *JHBPd2* [21]. Our study aimed to investigate the role of JHBPd2 in silk protein synthesis.

## 2. Results

### 2.1. Expression of BmJHBPd2 in Different Silk-Producing Strains

After many years of natural selection and artificial domestication, significant differences in silk production exist among different strains of silkworms. To analyze the mRNA levels of *BmJHBPd2* among different strains, the high-silk-producing practical strain 872 (*S872*) and low-silk-producing local strain *Dazao* were selected for further analysis to ascertain the differences between the strains. This study investigated the indicators of four main characteristics of *S872* and *Dazao*. The pupae and cocoons of *S872* were substantially larger than those of the *Dazao* strain (Appendix A). The whole cocoon weight, pupae, and cocoon weight, particularly the shell weight percentage of *S872*, were significantly higher than those of *Dazao* (Appendix A). Considering the high efficiency of silk protein synthesis by the silk glands of fifth instar larvae, the expression levels of genes in the PSG were investigated. In this study, we determined the expression levels of major silk protein genes using real-time quantitative reverse transcription polymerase chain reaction (qRT-PCR). *Bmfib-H*, *Bmfib-L*, and *BmP25* were specifically expressed in the PSG. The transcript levels of *Bmfib-H* and *Bmfib-L* were higher in the *S872* PSG than in that of *Dazao* at the third and fifth day of the fifth instar (Figure 1A,B). *BmP25* was expressed at the same level in both strains (Figure 1C). Differences in the *BmJHBPd2* expression were also analyzed. The transcript level of *BmJHBPd2* in the *S872* PSG was higher than that in *Dazao* on the third and fifth day of the fifth instar. The expression level of *BmJHBPd2 in S872* on the fifth day of the fifth instar was significantly higher than that in *Dazao* (Figure 1D). These results suggest that *BmJHBPd2* may be involved in silk protein synthesis.

### 2.2. Overexpression of BmJHBPd2 at the Cellular Level

JH regulates silk protein synthesis, and JHBP plays an important role in JH regulation as a vital response factor. To further explore how *BmJHBPd2* is involved in silk protein synthesis, the expression of the genes associated with the silk protein synthesis and those related to the JH signaling pathway in the exogenous JH induction were analyzed at the cellular level. First, the *BmJHBPd2* subcellular localization vector was constructed (Figure 2A). Immunofluorescence experiments showed that FLAG-tagged *BmJHBPd2* was localized in the cytoplasm (Figure 2B). This result is consistent with that reported by Li et al. [21] and implies that *BmJHBPd2* cannot be secreted and plays a physiological role in cells. The qRT-PCR analysis at the nucleic acid levels and Western blotting at the protein levels showed that the intracellular overexpression of BmJHBPd2 was successful (Figure 3A). BmE cells contain the signal transduction pathway for JH [30]. When adding JHA to BmE cells, the expression of *BmJHBPd2* was significantly upregulated. Furthermore, JHA significantly upregulated the expression of the early response factor *BmKr-h1*, suggesting that JHA significantly activated the JH downstream signaling pathway (Figure 3B). When only *BmJHBPd2* was overexpressed, *Bmkr-h1* was nearly unexpressed, and there was no change in the expression of the *B. mori* methoprene-tolerant 1 (*BmMet1)* gene (Figure 3C). However, the upregulated expression level of *BmKr-h1* in pSLfa1180-basic was higher than that in pSLfa1180-BmJHBPd2 (Figure 3D). These results infer that the excessive expression of BmJHBPd2 in the transfected cells reduced the amount of JH and then lowered the expression of *Bmkr-h1,* since the overexpression of BmJHBPd2 may be sufficiently high, as shown in Figure 3A, even though the expression of *Bmkr-h1* in Figure 3B increased. In addition, the expression of *Bmfib-H* was not detected in all experiments using BmE cells; this may have been caused by the lack of transcription factors specific to silk protein genes in the cells.

### 2.3. Transgenic Overexpression of BmJHBPd2 in the Silk Gland

To further explore the biological function of *BmJHBPd2* in the silk gland, a piggyBac transgenic vector containing a combination of *BmJHBPd2* and a Myc foreign label with the Fib-L promoter, Fib-L terminator, and *pBac [3xP3 EGFP]* was constructed (Figure 4A). The vector and helper plasmids were injected into 271 pre-blastoderm eggs, of which 136 hatched and developed to the adult stage. An EGFP-positive brood was obtained and used to establish the transgenic overexpression line (Figure 4B). Then, we obtained four positive G2 generations, and the results of the investigations on all four G2 generations showed that the synthesis of silk proteins was affected. One G3 generation was conserved and continued to be reared and investigated, and the result remained the same. The mRNA levels of *BmJHBPd2* in the PSG of larvae on the third day of the fifth instar larvae (L5D3) of the transgenic and wild-type (WT) lines were detected using qRT-PCR. The result showed that *BmJHBPd2* levels were substantially higher in the transgenic line than that in the WT line (Figure 4C). To confirm whether BmJHBPd2 with a Myc-tag was synthesized in the transgenic line, proteins were extracted from the PSG of L5D3 for Western blotting. The signals with the Myc antibody were only detected in the transgenic line (BmJHBPd2OE) but not in the WT line (Figure 4D). These results indicated that *BmJHBPd2* was overexpressed in the PSG.

### 2.4. Overexpression of BmJHBPd2 Affects Silk Synthesis and Silk Yield

This study investigated the strain overexpressing the *BmJHBPd2* gene obtained above, predominantly focusing on the silk gland and yield of fifth instar larvae. The BmJHBPd2OE line was raised to the L5D3 stage, and its silk glands were dissected and observed. There was no pronounced difference in the biological characteristics of the silk glands (Figure 5A). However, we found that the overexpression of *BmJHBPd2* resulted in thinner and smaller cocoon shells than those in the WT lines (Figure 5B). Further observation of the cocoon shells of the two lines showed that the whole cocoon weight, cocoon weight, pupae weight, and cocoon shell rate were significantly reduced in the BmJHBPd2OE line (Figure 5C–F). Based on these differences in cocoon shells between the BmJHBPd2OE line and WT line, we chose the L5D3 stage to determine the mRNA levels of silk protein-related transcription factors and silk fibroin genes in both lines. Among the silk fibroins tested, *Bmfib-H* and *Bmfib-L* were significantly downregulated at the L5D3 in the BmJHBPd2OE line (Figure 5G,H), except for *BmP25*, which did not differ significantly between the two lines (Figure 5I). Among the silk-related transcription factors, the expression of *Bmsage* and *Bmdimm* was significantly downregulated in the BmJHBPd2OE line (Figure 5J,K). However, the expression of *Bmsgf-1* was significantly upregulated (Figure 5L). These results indicate that this study successfully overexpressed the *BmJHBPd2* gene, although the expression of the silk fibroin gene and silk fibroin-related transcription factors in the BmJHBPd2OE line was significantly reduced, which affected the silk yield.

### 2.5. Overexpression of BmJHBPd2 Led to Repression of Silk Synthesis by Inhibiting Bmkr-h1 Expression in the Silk Glands

JHBPd2 affected the expression of *Kr-h1* in the JH pathway at the cellular level. Therefore, the major JH regulatory pathway genes were investigated in the silk glands of the BmJHBPd2OE and WT lines. According to qRT-PCR, the relative expression level of the early response factor, *BmKr-h1,* in the JH pathway was significantly reduced in the BmJHBPd2OE line (Figure 6A). JH receptors, such as Met1, methoprene-tolerant 2 (Met2), and steroid receptor co-activator (SRC), were also downregulated (Figure 6C,D). The key enzymes of the JH metabolic pathway, JH esterase (JHE), and JH epoxide hydrolase (JHEH), were significantly downregulated (Appendix A). Therefore, both the major genes in the JH pathway and the JH-degrading enzymes were downregulated, indicating that the JH pathway was affected by overexpression of *BmJHBPd2*.

*Kr-h1* can directly inhibit the biosynthesis of 20-hydroxyecdysone (20E) and the expression of some early transcription factors in 20E [31]. Therefore, the expression of some early transcription factors in 20E was investigated. This study found that the early transcription factor *B. mori* broad complex (*BmBrc*) of 20E was significantly upregulated (Figure 7A). The relative expression levels of E74A and ultraspiracle protein (USP) were significantly upregulated in the BmJHBPd2OE line (Figure 7B,C). Meanwhile, those of the ecdysteroid receptor (EcR), hormone receptor 3 (HR3), and E75A did not differ significantly between the two lines (Figure 7D–F). These results suggest that overexpression of *BmJHBPd2* increases the expression of early transcription factors in the 20E signaling pathway.

## 3. Discussion

The most valuable aspect of silkworm studies is the potential for increased production of silk [22]. In order to increase silk production, it is important to understand the process of silk protein synthesis [32]. In our study, we found a correlation between *BmJHBPd2* and silk protein gene expression and silk yield. At the cellular level, *BmJHBPd2* was induced by JHA and suppressed JH signaling by inhibiting the expression of *Bmkr-h1*. Individual experiments showed that overexpression of *BmJHBPd2* promoted the expression of 20E-related transcription factors by inhibiting the expression of *Bmkr-h1*, thereby decreasing the expression of silk protein genes and silk production. Our results indicate that *BmJHBPd2* plays an important role in regulating JH signaling in silk glands. Simply increasing the expression of JHPBd2 does not increase silk yield; rather, silk protein synthesis is inhibited. Our research provides a reference for future genetic modifications to improve silk yield.

As a specific carrier of the endocrine hormone JH in silkworms, JHBP protects and transports synthesized and secreted JH from the corpora allata [6]. Given that the organs are used for silk protein synthesis and secretion, silk glands grow rapidly during the fifth instar. Although JH has been found to be largely absent from the blood of fifth instar silkworms [33], to date, there have been no reports on whether the silk glands of fifth instar silkworms contain JH. Based on the functional studies, the findings of this study suggest that the silk glands of fifth instar silkworms are likely to contain JH and that JHBPd2 may play a role in regulating the concentration gradient of JH in the functioning of silk glands. The rationale is that the overexpression of JHBPd2 alone in the PSG increases its protein production, and JH entering the silk gland without reaching the JH concentration in the blood binds to the overexpressed JHBPd2. This can result in a decrease in free JH in the silk gland, which, in turn, reduces the expression of *Kr-h1*. As a key transcription factor connecting the JH and the 20E pathways, *Kr-h1* can directly inhibit the biosynthesis of 20E [31], thereby inhibiting insect growth. *Kr-h1* also directly binds to the Kr-h1 binding site (KBS) elements of E93, Brc, and E75 promoters to inhibit their expression [34,35,36,37]. During crosstalk between the JH and 20E pathways, *Kr-h1* is located upstream of the 20E pathway genes and inhibits their expression. The 20E transcription factors, such as Brc, strongly repress silk protein synthesis [38]. Consequently, the expression of silk proteins is reduced, which then leads to decreased silk yield.

The expression pattern of the *JHBPd2* gene is highly similar to that of the silk fibroin gene, both of which are highly expressed at the fifth instar stage and are mainly expressed in the PSG. This indicates a close relationship with silk fibroin synthesis [39]. The larval stage of silkworms was found to be positively correlated with the silk yield [40,41]. The expression of JHBPd2 was significantly higher in high-silk-yield varieties than in low-silk-yield varieties, and there was a positive correlation between its expression and silk yield. However, the overexpression of JHBPd2 was found to inhibit silk yield. This study also conducted an in-depth analysis of this issue. This contradictory results suggest that JHBPd2 plays a role in the regulation of JH concentration. The larval stage of high-silk-yield *S872*, especially at the end of the fifth instar, is 2–3 days longer than that of the low-silk-yield variety *Dazao*. This suggests that *S872* contains more JH than *Dazao* in vivo, which can be inferred from applying JH to the silkworm body surface, prolonging the developmental time of the silkworm [30]. With more JH in the high-silk-yield *S872*, there is a corresponding increase in JH content in the silk gland, which requires more JHBPd2 protein to bind and protect JH. Therefore, the silk glands of the high-silk-yield *S872* have more time to synthesize more silk protein and, thus, produce more silk. After the overexpression of BmJHBPd2, because the JH signaling pathway in the silkworm was affected, the balance of JH concentration in the silk gland was disrupted, which inhibited silk protein synthesis. Our results also indicate that simply changing a gene that is positively associated with silk production may not improve silk yield. Silk yield is a quantitative trait controlled by multiple genes. Varieties with high silk yields can result from artificial selection, which is the result of the synergistic regulation of multiple genes. Altering only one gene, such as JHBPd2 in this study, may disrupt homeostasis in vivo, which, in turn, inhibits silk protein synthesis. Therefore, further research is required to determine how to improve silk production through genetic manipulation. Further research on the regulatory mechanism of silk protein synthesis is expected to identify the most critical factors affecting silk protein synthesis.

In this study, we did not measure JH in the silk glands of the JHBPd2OE line due to limited material availability; however, we performed JH assays on normal silk gland tissues and found that silk glands contained JH. We propose the following regarding the expression of *BmJHBPd2* in the PSG (Figure 8). JH is transported into PSG cells from the hemolymph early in the fifth instar stage. Free JH then binds to the nuclear receptor *BmMet* and forms a complex with *BmSRC* [42]. This complex activates the expression of *BmKr-h1* and, subsequently, the expression of the transcription factor *Bmdimm* to regulate *Bmfib-H* [30]. Simultaneously, JH induces the upregulated expression of *BmJHBPd2* (Figure 3B). Cytoplasmic *BmJHBPd2* can bind to redundant JH and slowly release it to maintain the JH level, which continuously regulates gene expression for silk synthesis. Therefore, there are two potential sources of JH in the silk glands, one of which possibly originates from the blood. There is a consensus that JH is released into the blood after synthesis by the pharyngeal lateral body and that JH in the blood is bound by JHBP and transported to various tissues and organs. However, it is difficult to understand that at the fifth instar, JH is essentially undetectable in the blood. Therefore, it is unlikely that other tissues and organs contain JH. However, at the early age of the fifth instar, JH is likely synthesized by the corpora allata and then transported to other tissues and organs after being bound by JHBP in each organ. Here, JHBP functions as a sponge, which slowly releases JH and regulates the growth and development of each tissue and organ. The issue of the catch and release should be addressed in future research. However, the silk gland may synthesize JH independently. Although the corpora allata is the main JH-synthesizing organ, the possibility that other tissues and organs may synthesize JH cannot be excluded. In addition, a substantial number of JH synthesis enzymes have been detected in silk glands. Further confirmation of this is required.

## 4. Materials and Methods

### 4.1. Silkworm Strains and Cell Culture

*Bombyx mori*, the low-silk-producing strain *Dazao*, and the high-silk-producing strain *872* (*S872*), were provided by the State Key Laboratory of Silkworm Genome Biology, Southwest University in Chongqing, China. Silkworm eggs were cultured at a standard temperature of 25 °C under 12 h light and 12 h dark cycle conditions. The larvae were reared with fresh mulberry leaves with 75% relative humidity. The *B. mori* cell line, BmE [43], originally derived from embryo cells and was maintained at 27 °C in Grace’s medium supplemented with 10% FBS (HyClone, San Angelo, TX, USA).

### 4.2. RNA Preparation and Quantitative Real Time-PCR (qRT-PCR)

Total RNA was extracted from the cells and silk glands using TRIzol™ reagent (Invitrogen, Waltam, MA, USA). The GoScrip™ Reverse Transcription system (Promega Madison, WI, USA) was used for RT-PCR. The semiquantitative RT-PCR conditions were as follows: 95 °C for 10 s, followed by 25 cycles at 95 °C for 10 s, 55 °C for 15 s, 72 °C for 1 min, 72 °C for 7 min, and then maintained at 16 °C. Reverse transcription was performed using the PrimeScript™ RT reagent Kit with gDNA Eraser (Takara, Shiga, Japan). Quantitative PCR was performed using SYBR^®^ Premix Ex Taq™ II (Takara) and qPCR reaction under the following conditions: 95 °C for 10 s, followed by 40 cycles of treatment at 95 °C for 5 s, and 60 °C for 31 s. The silkworm ribosomal protein L3 (*BmRpl3*) was used as the internal marker gene. Three independent replicates were performed for each experiment.

### 4.3. Subcellular Localization

Primers for amplifying the ORF of *BmJHBPd2* are listed in Appendix A. Target fragments were obtained by gel purification and cloned into an pSLfa1180 (pSLfa1180-A4-EGFP-SV40) expression vector (maintained in our laboratory) between *BamH*I and *Not*I sites. Highly purified plasmid DNA was prepared using Qiagen Plasmid Midi kits (Qiagen, Dusseldorf, Germany). For the subcellular localization assay of silkworm BmJHBPd2, BmE cells were seeded onto coverslips in 24-well plates. After 12 h of culture, pSLfa1180-Basic, pSLfa1180-A4-EGFP, and pSLfa1180[A4-EGFP-BmJHBPd2-SV40] were separately transfected into BmE cells at 1 µg per well. Cells were transfected with expression plasmids using the X-tremeGENE HP DNA Transfection Reagent (Roche Applied Science, Penzberg, Germany). After transfection for 48 h, cells were fixed for 10 min at room temperature with 4% (*v*/*v*) formaldehyde in PBS. They were then blocked for 30 min in PBS containing 0.1% (*w*/*v*) BSA and 5% (*v*/*v*) goat serum. The samples were treated with a primary antibody (anti-FLAG monoclonal M2 mouse (Sigma-Aldrich, St. Louis, MO, USA)) for 1 h before being incubated with a secondary antibody (anti-mouse Alexa 488) for 30 min at room temperature. The samples were then mounted using a mounting medium containing 4,6-diamidino-2-phenylindole (DAPI) and photographed using a confocal microscope (FV1000; Olympus, Tokyo, Japan).

### 4.4. Western Blotting

Radio-immunoprecipitation assay (RIPA) lysis buffer (Beyotime, Shanghai, China) was used to extract proteins from the cells and the PSG. The lysate was divided evenly and then centrifuged for 5 min at 12,000× *g*. Protease inhibitors were then added to the supernatants. Protein concentrations were measured using a bicinchoninic acid (BCA) protein assay kit (Beyotime, Shanghai, China). The proteins were separated using 10% sodium dodecyl sulfate-polyacrylamide gel electrophoresis (SDS-PAGE) and transferred onto a polyvinylidene difluoride (PVDF) membrane (Roche, Basel, Switzerland). The PVDF membrane was blocked using 5% skimmed milk overnight at 4 °C and incubated with a primary antibody against BmJHBPd2 (1:10,000) for 2 h at 37 °C. After washing the PVDF membrane six times at 5 min intervals, the membranes were incubated with the secondary antibody goat anti-rabbit IgG (1: 20,000), labeled with horseradish peroxidase (HRP) (Sigma-Aldrich, St. Louis, MO, USA), and visualized with SuperSignal™ West Femto Maximum Sensitivity Substrate (Thermo Fisher Scientific, Waltham, USA) using the automatic exposing pattern of Genome XRQ (Gene Company, Hong Kong, China).

### 4.5. Statistical Method for Cocoon Layer Proportion

The whole cocoon was weighed and then gently peeled; following this, the pupa was removed, the epidermis was shed, and the remaining cocoon was then weighed again. The ratio of this weight to the whole cocoon weight was determined as the cocoon layer ratio, which was measured in each of 15 individuals of transgenic and wild-type silkworms of *Dazao*.

### 4.6. Plasmid Construction to Obtain Transgenic Silkworms

To construct the transgenic overexpression lines, we used the *pBac [3xp3-EGFP]* system. JHBPd2 was driven by the PSG-specific Fib-L promoter to obtain the *pBac [3xp3-EGFP-Fib-L-BmJHBPd2-myc]* (Figure 4A). The target gene, *BmJHBPd2*, was amplified by PCR using cDNA from the silk gland tissue of the *Dazao* cultivar, using that of the third day of the fifth instar as the template. The 5’ end of *BmJHBPd2*-F was selected to add the *BamH* I (Takara, Tokyo, Japan) restriction endonuclease site. The 5’ end of *BmJHBPd2*-R was selected to add the *Not*I (Takara) restriction endonuclease site and Myc foreign label sequence for amplification. Full-length PCR products were digested with *BamH*I and *Not*I and connected to the 1180 vector skeleton with the Fib-L promoter to construct the *psl1180 [FibL-BmJHBPd2-myc]* expression vector. The *Asc*I (Takara) enzyme was used to digest the *psl1180 [FibL-BmJHBPd2-myc]* carrier and *pBac [3xp3-EGFP]* vector. Solution I (Takara) was used to construct an overexpression vector. The primers used to construct the plasmids are listed in Appendix A.

### 4.7. Silkworm Germline Transformation

For silkworm germline transformation, the ultrapure plasmid of *BmJHBPd2* overexpression transgenic vector and helper plasmid were mixed at a 1:1 volume ratio, and the final concentration for embryo injection was 300–500 ng/μL. After being sealed, they were moved into an artificial culture room and incubated at 25 °C for 10–12 days until the larvae hatched. After hatching by injecting the silkworm eggs, the G0 generation could not be screened for transgenic individuals [44]. Males and females were randomly mated and laid eggs for the G1 generation. The eggs of the G1 generation eggs were fluorescently screened at around the sixth day of development. Egg circles with eyes emitting green fluorescence were screened as positive individual egg circles and raised to produce the next generation. Homozygous third-generation transgenic silkworms were used for molecular-level detection in subsequent experiments.

### 4.8. Statistical Analysis

All the data were statistically analyzed using Student’s *t*-tests. Asterisks indicate significant differences (* *p* < 0.05, ** *p* <0.01, and *** *p* < 0.001). ns is not statistically different.

### 4.9. JHA Treatment

BmE cells were seeded onto coverslips in 24-well plates. After 12 h of culture, pSLfa1180-Basic and pSLfa1180-BmJHBPd2 were separately transfected into BmE cells at 1 µg per well. After transfection for 8 h, the medium was changed. JHA was melted in DMSO [30]. We then added 1 ng JHA (Sigma-Aldrich, Methoprene, 40596-69-8) to each well, and the cell RNA was extracted after hormone treatment for 12 h.

## Figures and Tables

**Figure 1 ijms-24-12650-f001:**
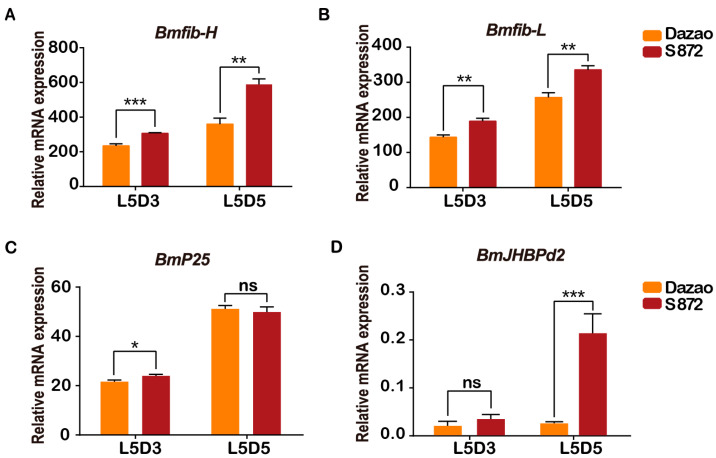
The expression of *BmJHBPd2* was different in the PSG among the two silk-producing strains *Dazao* and *S872*. (**A**–**D**). Relative *Bmfib-H*, *Bmfib-L*, *BmP25*, and *BmJHBPd2* mRNA levels in the PSG of *Dazao* and *S872* larvae on the 3rd and 5th day of the fifth instar as analyzed by real-time quantitative reverse transcription polymerase chain reaction (qRT-PCR). *BmRpl3* was used as an internal control. The results are expressed as means ± standard deviation (SD) of three independent experiments; * *p* < 0.05; ** *p* < 0.01; *** *p* < 0.001; ns, not significant.

**Figure 2 ijms-24-12650-f002:**
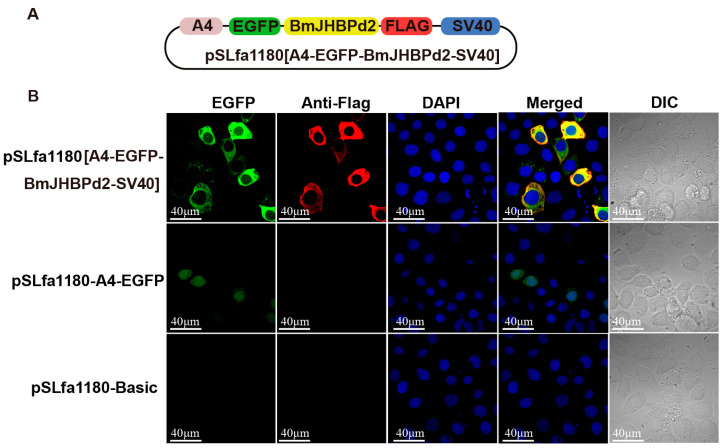
Overexpression of *BmJHBPd2* in BmE cells. (**A**). Structural map of subcellular-localized overexpression of the *BmJHBPd2* vector. (**B**). Immunofluorescence experiment of *BmJHBPd2* in BmE cells.

**Figure 3 ijms-24-12650-f003:**
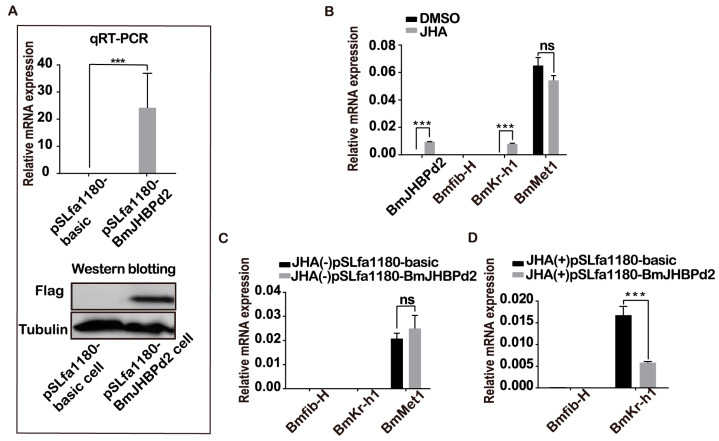
*BmJHBPd2* responds to juvenile hormone analogs (JHAs) in BmE cells. (**A**). Overexpression of *BmJHBPd2* in BmE cells assayed by qRT-PCR and Western blotting using the flag tag antibody and tubulin as a control. (**B**). Expression of *BmJHBPd2*, *Bmfib-H*, *BmKr-h1,* and *BmMet1* after adding JHA or DMSO to normal BmE cells. (**C**). Expression of *Bmfib-H*, *BmKr-h1,* and *BmMet1* after overexpression of *BmJHBPd2* assayed without adding JHA. (**D**). *Bmfib-H* and *BmKr-h1* expression level in BmE cells overexpressing *BmJHBPd2* after adding JHA. The experiments of (**B**–**D**) were assayed using qRT-PCR, and *BmRpl3* expression was used as an internal control. Results are expressed as the means ± S.D. of three independent experiments. *** *p* < 0.001; ns, not significant.

**Figure 4 ijms-24-12650-f004:**
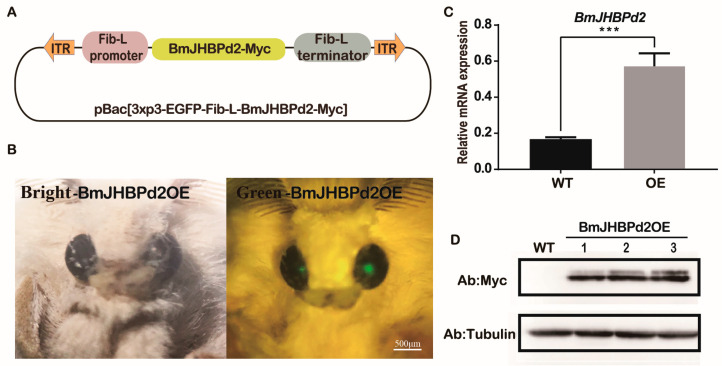
Overexpression of *BmJHBPd2* in the posterior silk gland. (**A**). Schematic diagram of carrier construction. (**B**). Screening of transgenic moths; transgenic-positive individual moths under white light and green fluorescence. (**C**). Overexpression of *BmJHBPd2* assayed using qRT-PCR; *BmRpl3* was used as an internal control. (**D**). Detection of BmJHBPd2 at the protein level using a Myc tag antibody and tubulin as a control. Results are expressed as means ± S.D. of three independent experiments. *** *p* < 0.001.

**Figure 5 ijms-24-12650-f005:**
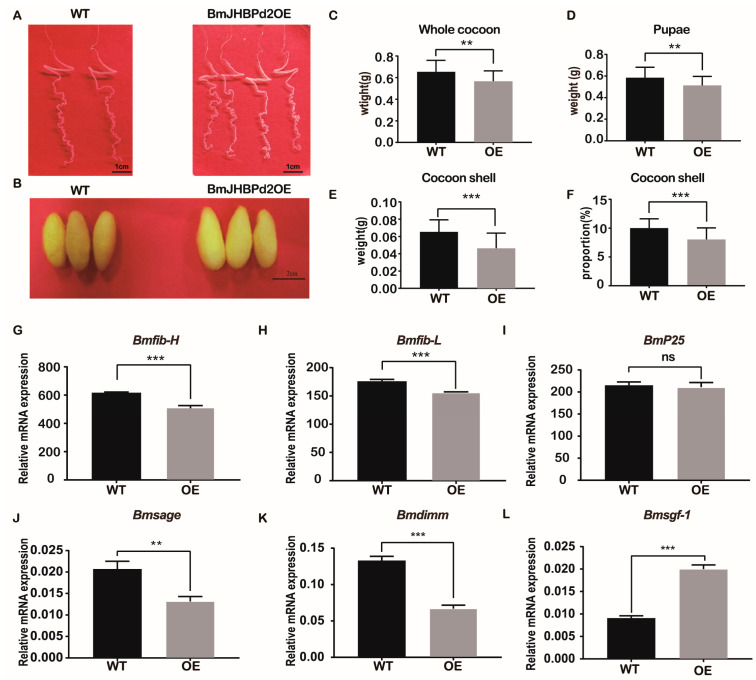
Analysis of fibroin gene expression and cocoon traits. (**A**–**F**). Phenotype of silk gland at the fifth instar and cocoon of the BmJHBPd2OE and WT lines. (**G**–**L**). Expression of silk fibroin protein gene and silk protein transcription factor in the WT and transgenic lines. *BmRpl3* expression was used as a control. Results are expressed as means ± S.D. of three independent experiments. ** *p* < 0.01; *** *p* < 0.001; ns, not significant.

**Figure 6 ijms-24-12650-f006:**
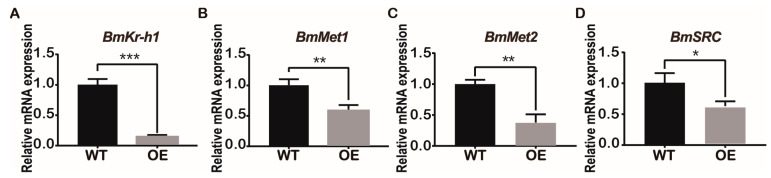
Overexpression of *BmJHBPd2* results in the inhibition of JH signaling pathway in silk glands assayed by qRT-PCR. The following JH signaling pathway–related genes were selected: Kr-h1 (**A**), Met1 (**B**), Met2 (**C**), and SRC (**D**). *BmRpl3* was used as a control. Results are expressed as means ± S.D. of three independent experiments. * *p* < 0.05; ** *p* <0.01; *** *p* < 0.001.

**Figure 7 ijms-24-12650-f007:**
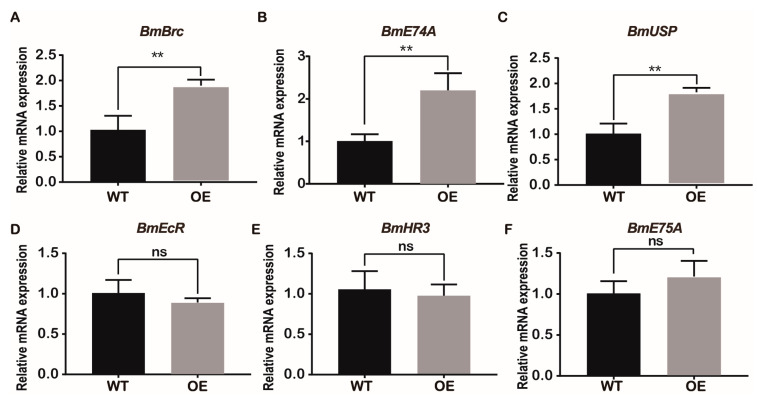
Overexpression of *BmJHBPd2* results in upregulation of the 20-hydroxyecdysone (20E) signaling pathway in silk glands assayed using qRT-PCR. The following 20E signaling pathway–related genes were selected: Brc (**A**), E74A (**B**), USP (**C**), EcR (**D**), HR3 (**E**), and E75A (**F**). *BmRpl3* was used as a control. Results are expressed as means ± S.D. of three independent experiments. ** *p* < 0.01; ns, not significant.

**Figure 8 ijms-24-12650-f008:**
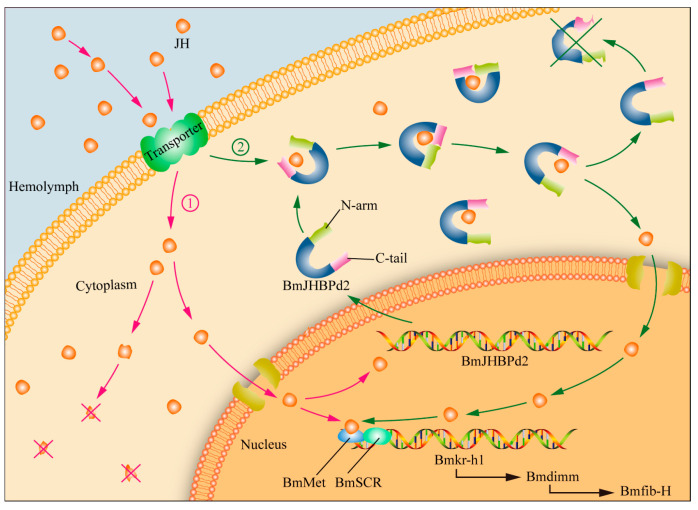
Function prediction model of BmJHBPd2 in the PSG. JH is transported into PSG cells from the hemolymph and functions in two ways. ①: Free JH binds to the nuclear receptor *BmMet* and forms a complex with *BmSRC*. ×: Free JH degradation in cytoplasm. ②: Cytoplasmic BmJHBPd2 can bind to redundant JH and slowly release it to maintain the JH level. And then, JH enters the nucleus and binds to the nuclear receptor *BmMet* and forms a complex with *BmSRC*.

## Data Availability

The data presented in this study are available on request from the corresponding author.

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
